# Effect of Temperatures on Polyphenols during Extraction

**Anila Antony and Mohammed Farid ***

Department of Chemical and Materials Engineering, University of Auckland, Private Bag 92019, Auckland 1010, New Zealand; aant520@aucklanduni.ac.nz
* Correspondence: m.farid@auckland.ac.nz

**Abstract:** Background: Polyphenols are a set of bioactive compounds commonly found in plants. These compounds are of great interest, as they have shown high antioxidant power and are correlated to many health benefits. Hence, traditional methods of extraction such as solvent extraction, Soxhlet extraction and novel extraction technologies such as ultrasound-assisted extraction and subcritical water extraction (SWE) have been investigated for the extraction of polyphenols. Scope and Approach: Generally, for traditional extractions, the total phenolic content (TPC) is highest at an extraction temperature of 60–80 °C. For this reason, polyphenols are regularly regarded as heat-labile compounds. However, in many studies that investigated the optimal temperature for subcritical water extraction (SWE), temperatures as high as 100–200 °C have been reported. These SWE extractions showed extremely high yields and antioxidant capacities at these temperatures. This paper aimed to examine the relevant literature to identify and understand the mechanisms behind this discrepancy. Results: Thermal degradation is the most common explanation for the degradation of polyphenols. This may be the case for specific or sub-groups of phenolic acids. The different extraction temperatures may have also impacted the types of polyphenols extracted. At high extraction temperatures, the formation of new compounds known as Maillard reaction products may also influence the extracted polyphenols. The selection of source material for extraction, i.e., the plant matrix, and the effect of extraction conditions, i.e., oxidation and light exposure, are also discussed. The overestimation of total phenolic content by the Folin–Ciocâlteu assay is also discussed. There is also a lack of consensus in TPC's correlation to antioxidant activity.

**Keywords:** polyphenols; antioxidant; bioactive compounds; pressurised liquid extraction; subcritical water extraction; solvent extraction; thermal degradation



## 1. Introduction

Polyphenols are naturally occurring substances found primarily in fruits, vegetables and most plant materials. Polyphenol is an umbrella term, so phenolic acids, flavonoids, stilbenes and lignans are all types of polyphenols. These substances show high antioxidant activity and are correlated to many positive health benefits [1,2]. Polyphenols are attributed with having protective effects against diabetes, cancer and cardiovascular diseases (CVD) [3]. The most well-studied subclass of polyphenols are flavonoids [4]. Some of the most important flavonoids are catechins, hesperidin, naringenin and quercetin. While the exact mechanism behind how these substances provide these health benefits is not well understood, studies show a clear positive correlation. Many of these positive benefits have been attributed to their antioxidant capacity, and the global market for antioxidants has grown to about US$3000 million [5]. With such a rise, there has recently been a growing number of studies investigating various extraction techniques.

The extraction techniques can be broadly divided into traditional (conventional) and novel (emerging) technologies. The traditional extractions include solvent extraction, Soxhlet extraction and maceration. The disadvantages of these extractions are the long extraction times, high energy consumption, the need for expensive, high-quality organic solvents, evaporation of the solvents during treatment and the use of these potentially toxic

solvents, which then poses problems with disposal and possible loss of functionality [6]. The novel technologies include ultrasound-assisted extraction (UAE), enzymatic extraction, microwave-assisted extraction (MAE) and pulsed electric field (PEF) extraction. These emerging technologies are non-thermal, do not use any hazardous chemicals, provide a shorter extraction time and have high energy and extraction efficiency. Most of these novel extraction technologies are considered "green", as they fit the criteria set by the Environmental Protection Agency (EPA) [7,8].

Supercritical/subcritical fluid extraction is another prominent technique in the field. This technology is often categorised as both a novel and traditional technology depending on the individual user. However, unlike other emerging technologies, this process does sometimes include treatment at high temperatures. The fluid chosen is also of importance for categorisation, as the critical temperature of the different fluids varies widely. In the case of supercritical $CO_2$ extraction, the treatment may be considered non-thermal (around 40–60 °C); for example, the critical point for $CO_2$ is around 31.2 °C [9]. As this technique may also use propane, methanol and other chemicals as solvents, it is sometimes categorised as a traditional extraction technology. On the other hand, subcritical water extraction is known by many names—pressurised liquid extraction (PLE), pressurised hot water extraction (PHWE) and superheated water extraction (SHWE). As water is a low-cost, non-toxic and environmentally friendly solvent, this extraction technique is very beneficial [10]. However, pressurised liquid extraction (hereafter referred to as PLE) often requires high temperatures between 100 °C and 300 °C and, hence, a very high-pressure vessel for treatment.

In traditional extractions, the highest polyphenol yield is observed around 60–80 °C, and the most common solvents used for these extractions are ethanol, methanol and acetone [10–27]. As there is solvent loss at temperatures above 60–80 °C, studies investigating the impact of extraction temperatures did not include higher treatment temperatures (>80–90 °C). Studies that investigated the impact of drying temperature on polyphenolic yields have provided insight into the behaviour of polyphenols at these temperatures. Many studies have shown that at drying temperatures above 80 °C, the polyphenolic yield decreases [19,28–35]. For this reason, polyphenols are regularly regarded as heat-labile compounds.

However, pressurised liquid extraction (PLE) studies have shown high phenolic yield at temperatures well above 100 °C. Generally, an increase in extraction temperature (up to 180–200 °C) correlates to an increase in phenolic content and antioxidant activity [36–53]. As this conflicts with findings from studies using traditional extractions, this topic requires further investigation. The aim of this paper was to examine the variety of complex mechanisms and factors that could explain the variance in the behaviour of polyphenols at high temperatures.

Note:
PLE: Pressurised liquid extraction, subcritical water extraction, hot water extraction
TSE: Traditional solvent extraction
TPC: Total phenolic content
TFC: Total flavonoid content
AOA: Antioxidant activity

## 2. Thermal Degradation

Maillard and Berset [54] used three mechanisms to explain the behaviour of polyphenols at high temperatures. First, the insoluble phenolic compounds may be released when the lignin bonds to phenolic acids are broken. It has been shown that the quantity of bound phenolic acids (measured after hydrolysis of plant tissue) is twice that of free phenolic compounds [54]. Secondly, lignin itself may be degraded at high temperatures, giving rise to more phenolic acids. This could explain the increase in phenolic yield with increasing temperature in PLE extractions. Lastly, at high temperatures, thermal degradation of the polyphenols may occur. Thermal degradation is the most common mechanism used to explain the fall in polyphenol yield during high-temperature extractions. However, ther-

mal degradation alone does not explain the behaviour of polyphenols, as conventional extraction studies have shown that thermal degradation occurs at 80 °C compared with 150–200 °C in PLE. The other two mechanisms discussed above could be attributed to the conflicting results in the literature. The increase in TPC observed at high temperatures in PLE extractions could be due to the lignin–phenolic acid bonds breaking or due to the breakdown of lignin itself, giving rise to more phenolic acid.

Larrauri et al. [35] reported a significant drop in total phenolic content when the plant material was dried at temperatures above 100 °C, also showing a decrease in antioxidant activity along with a decrease in TPC and suggesting that thermal degradation of the polyphenols may have occurred. Another study was conducted by the same authors to investigate the effect of drying temperatures between 20 °C and 120 °C. They observed that antioxidant activity at 20 °C was 1.7 times higher than the AOA observed at 120 °C. They suggested that the oxidation of polyphenols at high temperatures may also lead to a loss in observed antioxidant activity [28].

Ross et al. [34] studied the effect of heating grape seed flour at various temperatures (120–240 °C) on polyphenols extracted with ethanol. The study found that the TPC yield dropped when the temperature was raised beyond 180 °C, while total flavonoid content (TFC) dropped at a lower temperature of 120 °C, suggesting that flavonoids are more sensitive to temperatures. A significant increase in TPC was observed at 150 °C, which could be due to the liberation of phenolic compounds from the plant matrix and/or the breakdown of lignin. At an extremely high temperature of 180 °C, similar to Larrauri et al. [35], the authors suggested that the main mechanism behind the reduction is thermal degradation. When yields of specific polyphenols are calculated by HPLC, it is observed that (1) catechin drops at above 150 °C, (2) gallocatechin increases massively with increasing time and temperature, (3) epicatechin is not highly affected at 120 °C but drops at above 150 °C and (4) gallic acid increases massively with time and temperature [34]. Other studies have also found that flavonoids are more heat sensitive, and the degradation of flavones and flavanols/flavanones are observed at lower temperatures [39,40,54–59].

Anthocyanins are one of the most abundant flavonoid constituents found in fruits and vegetables [60,61]. One study tested the thermal effects between 20–60 °C and observed that, initially, an increase in temperature causes an increase in anthocyanin extraction [16]; however, a sharp decrease in anthocyanin extraction was observed at 45 °C. The anthocyanin content in the TPC dropped from 70% to 54% with an increase in temperature beyond 45 °C. The initial increase in yield with temperature could be due to the increase in solubility of anthocyanins in ethanol. However, the TPC showed the opposite trend and increased with temperature. The contrast in the yield of TPC and anthocyanins could be due to anthocyanins being more sensitive to high temperatures. Ju and Howard [62] investigated the impact of temperature on both anthocyanins and TPC. They also found that the optimal extraction temperature for anthocyanins occurred at lower temperatures (80–100 °C) compared with TPC (120 °C). Volden et al. [63] found that blanching (94–96 °C, 3 min) and boiling (10 min) red cabbage increased the total polyphenols while the anthocyanin content decreased by 59% and 41% for the blanched and boiled red cabbage, respectively, suggesting that the anthocyanins were degraded by heat. Another study found that the anthocynanin content of elderberry, strawberry and black carrot reduced with heating at 95 °C for 1 h. The anthocyanin content in elderberry decreased by 50% after 3 h of heating [64].

Table 1 provides a summary of the findings of the papers discussed above.

**Table 1.** Thermal Degradation.

| Source | Extraction | Temperatures Tested | Effect on Polyphenols | Reference |
|---|---|---|---|---|
| Red grape pomace peels | TSE | Dried at 60, 100 and 140 °C; freeze-dried samples served as controls | TPC ↓ at 100 °C | [35] |
| Grape seed flour (GSF) | TSE | Heated at 120, 150, 180, 210 or 240 °C | TPC ↓ above 180 °C<br>TFC ↓ above 120 °C | [34] |
| Black rice | TSE | Dried at 20, 40, 60, 80 and 100 °C | TFC ↓ above 40 °C<br>TPC ↓ above 80 °C | [30] |
| Spinach | PLE | Extractions between 50–190 °C | Flavonoids ↓ at 130 °C<br>No decrease in TPC | [41] |
| Black currants | TSE | Extractions between 20–60 °C | T ↑, TPC ↑<br>Anthocyanins ↓ above 45 °C. | [16] |
| Hemp, flax, canola seed cakes | TSE | Extractions at 40, 50, 60, 70 °C | T ↑, TPC ↑<br>TFC ↓ above 60 °C in flax and canola seed cake<br>TFC ↓ above 70 °C in hempseed cake | [57] |
| Peach | TSE | Extractions between 25–70 °C | TFC ↓ above 60 °C<br>TPC remains same between 25–70 °C | [55] |
| Mango peels and seed | TSE | Extractions at 25, 50, and 75 °C | TFC ↓ at 50 and 75 °C | [58] |
| Red grape skin | PLE | Extractions between 20 to 140 °C | Anthocyanins ↓ above 100 °C<br>TPC ↓ above 120 °C | [62] |
| Elderberry, strawberry and black carrot | TSE | Heated at 95 °C | Anthocyanins ↓ | [64] |
| Red cabbage | TSE | Blanched at 94–96 °C | TPC ↑ at 94–96 °C<br>Anthocyanins ↓ at 94–96 °C | [63] |

## 3. Effects of Other Parameters on Traditional and PLE Extractions, i.e., Oxidation, Light Sensitivity, Heating Time and Enzymes

### 3.1. Extraction Conditions

The duration of heating, storage conditions prior to extraction and exposure to oxygen are all factors that play an important role in determining the effect of temperature on the polyphenols [63,65]. PLE and traditional extraction are performed in very different environments. PLE is performed inside a cell with no exposure to light, mostly in the absence of oxygen (usually the cell is purged with nitrogen) for a shorter period compared with traditional extraction performed for long hours in a beaker. Long extraction times leave the partially extracted polyphenols exposed to light and oxygen, which has been reported to degrade the extractants [36]. This is further supported by studies conducted on the impact of various methods used for drying the plant material before extraction. Chan et al. [66] found that sun-drying ginger leaves caused significant loss in polyphenol yield compared with other drying methods that do not expose the leaves to light. Hot air or conventional drying has been shown to significantly decrease polyphenol yield compared with drying in a vacuum or nitrogen [30,67,68].

Ibañez et al. [37] performed a PLE extraction of polyphenols from dried rosemary leaves at 100, 150 and 200 °C. The study also found that when the temperature increased, the TPC increased. As the TPC and AOA were maximum at 200 °C, the authors suggested that thermal decomposition did not occur. As deoxygenated water was used and no exposure to air occurred during the process, it was suggested that a low level of oxidation occurred. Carnosic acid is said to be more sensitive to the presence of oxygen than temperature, which could explain why a PLE at 200 °C showed the highest yield of carnosic acid.

Palma et al. [36] studied the stability of phenolic compounds, specifically the impact of temperature and other conditions on individual polyphenol standards (gallic acid, cis-coumaric acid, caftaric acid, catechin, epicatechin and epigallocatechin gallate). The samples were subject to a superheated methanol treatment at 40, 50, 100 and 150 °C, while methanol extraction at 65 °C (boiling point of methanol) for 45 min was conducted as a control. The recovery rates of individual polyphenols were calculated using HPLC.

Ninety per cent of all phenolic compounds were recovered at all temperatures except for catechin and epicatechin, which saw a drop in recovery rate above 150 °C. The results of the boiling methanol extraction showed that the recovery rates were much lower than the PLE extraction. In fact, catechin saw a reduction of 40% in its concentration.

The highest degradation observed in superheated methanol treatment was for catechins, but it was only 14%. The authors suggested that this degradation happened at moderate temperatures (around 65 °C) during boiling methanol extraction due to light exposure and oxidation by the oxygen present in the air. As no such degradation was observed in dark and oxygen-free environments during superheated methanol extraction, it was concluded that the lower recovery rates in methanol extraction were not due to thermal degradation. As catechins are known to be the most oxidisable compound among the ones tested, the 40% loss of catechins in methanol extraction compared with only 10% in PLE provides more evidence that the degradation is due to the presence of oxygen.

### 3.2. Enzymes

Enzymes such as glycosidases, polyphenol oxidases (PPO) and peroxidases found in plant tissue can cause degradation of polyphenols [61]. Glycosidases can break down anthocyanins to anthocyanidins and sugars that are chemically unstable and rapidly degrade [61]. In the presence of oxygen, PPO can catalyse the oxidation of o-dihydrophenols diphenols (e.g., chlorogenic acid) into o-quinone (e.g., chlorogenoquinone) [60]. In the presence of $H_2O_2$, peroxidase can catalyse the same reaction [69].

The thermal stability of this enzyme is heavily influenced by the source of the enzyme but also depends on the pH [70]. Even PPO from the same source may have different thermostabilities depending on the different molecular forms of PPO [70]. However, in general, PPO is not considered an extremely heat-stable enzyme. PPO can be inactivated with mild heating, such as a blanching treatment for fruits and vegetables [60]. In most cases, a temperature of 70–90 °C inactivates the enzyme [70].

Rossi et al. [71] found that blanching blueberries led to the doubling of anthocyanin content in its juice compared to unblanched samples. The blanching treatment completely inactivated the native PPO and thus increased the anthocyanin recovery [71]. Additionally, another study found that the addition of unblanched blueberry-pulp extract to blueberry juice caused a 50% loss in anthocyanins while the addition of blanched extract did not result in any degradation [72].

### 3.3. Solvent Concentration and pH

The solvent concentration also has a significant impact on the optimal extraction temperature [16,41]. Cacace and Mazza [16] studied the impact of ethanol concentration on the optimal extraction temperature of polyphenols from blackcurrants. They found that extraction could be performed at lower temperatures when low ethanol concentration was used. For example, at 85% ethanol concentration (wt basis), the maximum anthocyanin extraction is between 30–35 °C, while at 20% ethanol (wt basis), the highest anthocyanin yield was observed at 25 °C.

The pH of the extraction media also plays an important role in determining the effect of temperature. Havlíková and Míková [65] investigated the stability of anthocyanins under various pH and temperatures. At lower temperatures (50–60 °C), pH plays a significant role in anthocyanin's thermal stability, but at temperatures above 70 °C, the pH does not have a significant effect. The pH during extraction did not affect the stability of the anthocyanins when the oxygen concentration during treatment was negligible.

## 4. Effects of Extraction Temperature on the Profiles of Polyphenols Extracted

Many studies have found that the extraction temperature significantly impacts the type of polyphenols extracted since various polyphenols degrade at different temperatures [36, 39,41,47,51,55,58,62,73–75].

Vergara-Salinas et al. [47] performed pressurised liquid extraction (PLE) on dried thyme and studied the effect of temperatures between 50 and 200 °C and the profile of polyphenols being extracted at each temperature using HPLC. The results showed that temperature had a significant effect on the polyphenol subclasses (Figure 1, Table 2). Hydroxyphenylpronanoic acid (HPPA) concentration increased by almost three times when the extraction temperature was 200 °C, while 100 °C was optimal for hydroxycinnamic acids, flavones, flavonols and total polyphenols. Additionally, higher extraction temperatures showed less diversity in the types of polyphenols extracted.

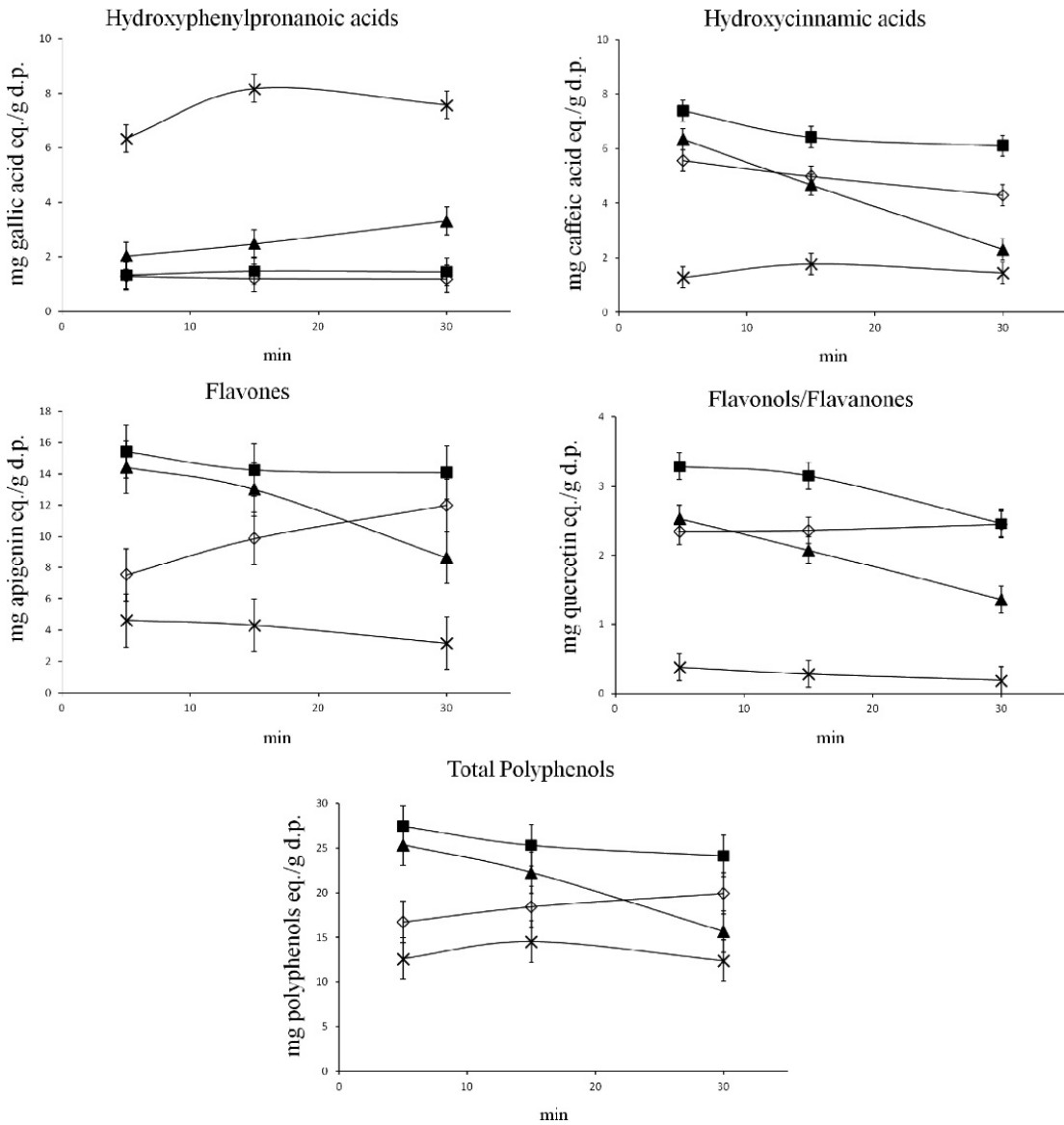

**Figure 1.** Impact of temperature on polyphenols and subclasses. Extraction temperatures: (◇) 50 °C; (■) 100 °C; (▲) 150 °C; (×) 200 °C. Reprinted with permission from {Vergara-Salinas, J.R.; Pérez-Jiménez, J.; Torres, J.L.; Agosin, E.; Pérez-Correa, J.R. Effects of temperature and time on polyphenolic content and antioxidant activity in the pressurized hot water extraction of deodorized thyme (*Thymus vulgaris*). J. Agric. Food Chem. 2012, 60, 10920–10929, doi:10.1021/jf3027759.}. Copyright {2012} American Chemical Society.

**Table 2.** Impact on individual phenolic compounds.

| Source | Extraction | Effect | Reference |
| --- | --- | --- | --- |
| Thyme | PLE | T ↑, hydroxyphenyl propanoic acids (HPPA) ↑, hydroxycinnamic acids ↓, flavones ↓, and flavanols ↓ | [47] |
| Rosemary | PLE | T ↑, rosmanol ↓, carnosol ↓, carnosic acid ↑ | [37] |
| Mint | TSE followed by acid hydrolysis | All phenolic increased massively after hydrolysis, except caffeic acid | [73] |
| Buckthorn | TSE followed by acid hydrolysis | Ferulic acid, myricetin, quercetin, naringenin, luteolin and apigenin appeared after hydrolysis. The content of the phenolics also increased with the exception of gallic acid, which slightly decreased. Vanillic acid was present in the normal extract but not present after hydrolysis | [73] |
| Birch | TSE followed by acid hydrolysis | Myricetin, quercetin and kaempferol appeared after hydrolysis. Hydrolysis caused an increase in the content of gallic acid, protocatechuic acid and apigenin but a decrease in the content of caffeic acid and chlorogenic acid | [73] |
| Caraway | TSE followed by acid hydrolysis | Protocatechuic acid was found in the normal extract but not found after hydrolysis. Caffeic acid decreased after hydrolysis | [73] |
| Parsley | TSE followed by acid hydrolysis | Quercetin appeared after hydrolysis. Increase in concentrations of all other phenolics after hydrolysis. Increase in gallic acid again explained by hydrolysis of galotannins | [73] |

Ibañez et al. [37] performed a sequential extraction at 100, 150 and 200 °C and analysed the extracts using HPLC; they found that polar phenolic compounds were extracted at low temperatures while less polar phenolics were extracted at higher temperatures. The polarity of water was reduced at higher temperatures, allowing it to solvate the nonpolar compounds and extract them. As a result, depending on the extraction temperature, phenolic compounds can be extracted with high selectivity. Hossain et al. [45] studied the extraction of polyphenols from rosemary, oregano and marjoram and found that TPC increased as the temperature increased from 66 °C to 200 °C. However, at temperatures above 150 °C, rosmarinic acid and carnosic acid decreases (Table 2). As the AOA remained very high, the authors suggested that rosmarinic and carnosic acid broke down to compounds with higher antioxidant power.

Palma et al. [36] performed a PLE for polyphenols from grape peels and seeds. For grape seed extracts, compounds were detected at 150 °C that were not present at 50 °C and 100 °C. Mišan et al. [73] extracted polyphenols from parsley, buckthorn, mint, caraway and birch and also performed acid hydrolysis (HCL in 50% aqueous methanol heated to 90 °C for 2 h) and compared the results to unhydrolysed samples. With the assumption that hydrolysis of the phenols occurs at high temperatures, we can compare the results of this study to other papers performing extractions at high temperatures. The results show that depending on the plant, many polyphenols appear or disappear after hydrolysis. Table 2 below summarises studies that showed a shifting profile of phenolics depending on extraction conditions.

## 5. Formation of Maillard Reaction Products (MRP) at High PLE Temperatures

There is limited understanding of the composition, quantity and function of new compounds that are formed under PLE extractions at high temperatures. These compounds are commonly referred to as Maillard reaction products or MRPs. They have been reported to possess antioxidant activity and be toxic, mutagenic compounds [32,76,77]. Plaza et al. [78] found that newly formed compounds possessed AOA when extracted using PLE at 200 °C. Hossain et al. [45] showed that MRPs increased when the temperature increased from

150 °C to 200 °C, as did the TPC and AOA. With extraction temperatures above 150 °C, studies have reported the formation of MRPs and an increase in AOA [45,50,78,79]. Due to the possible toxic effects from MRPs, extraction products obtained at these temperatures should be carefully analysed and studied.

## 6. Variance in Source

It is also important to note that a plant's phenolic content itself may vary depending on plant growing conditions and plant genotypes [73]. Depending on the species, various forms of polyphenols and how they are bound to the plant tissue may vary and generate different effects on extraction temperature [57]. Heat treatment could aid in breaking the phenol–protein and phenol–polysaccharide bonds that increase extraction yield [22]. As the different phenolics in plant tissue are bound differently, the most effective method to extract the phenolic compounds will be different based on the plant species [17].

Palma et al. [36] performed a PLE for polyphenols from grape peels and seeds. They found that the temperature of extraction did not have a significant impact on the recovery rates from grape skin, but it did from the grape seed. As the breaking of bonds between the phenols and the plant matrix was facilitated at high temperatures, the authors suggested that the phenolic compounds in grape seed must have stronger bonds to the matrix compared with grape skin.

Barros et al. [50] studied the impact of PLE temperature on two types of sorghum brans and found that the optimum temperature was different for each type. As the profile of polyphenols within each type of sorghum bran is different, the optimal extraction temperature is also different. As seen in Table 2 above, the phenolic profile of each plant is very different, and due to this diversity, the hydrolysis that occurs during extractions at high temperatures may explain the variances reported in the literature.

To better understand the impact of the source's plant matrix, we can specifically examine one well-studied source of polyphenols: honey. Many studies have investigated the impact of thermal processing on various types of honey, giving us insight into the behaviour of polyphenols.

Majkut et al. [80] found that among four nectar honey variants tested, all four showed an increase in TPC and AOA with thermal treatment at 100 °C. However, the extent to which it increased depended on the honey variant. For example, The TPC in rapeseed honey increased 15%, while a 27% increase was observed for buckwheat honey [80]. In contrast, Villacrés-Granda et al. [81] found that heat treatment at 60 °C caused a two-fold reduction in the TPC of eucalyptus honey.

Wang et al. [82] investigated the impact of thermal processing (82.2 °C for 10–12 s) on clover and buckwheat honey. There was no significant change in the TPC of clover honey, while the TPC in buckwheat honey showed a decrease. Another study [83] investigated the impact of thermal processing (90 °C up to 60 min) on honeydew, lime, acacia and buckwheat honey. The results, once again, varied depending on the origin of the honey. There was no significant change in TPC for acacia and buckwheat honey, while TPC increased for lime honey and decreased for honeydew honey [83]. Aydogan-Coskun, Coklar and Akbulut [84] compared the impact of liquefaction at 55 °C for 12 h and pasteurisation at 90 °C for 15 s on astragalus and sunflower–cornflower honey. They concluded that the variation in the impact of the process on TPC and AOA is based on the type of honey. Escriche et al. [85] studied the influence of heat treatments on phenolic profiles of citrus, rosemary, polyfloral and honeydew honey and also concluded that the flavonoids reacted differently to the heat treatments depending on the origin of the honey.

From the results of all the studies presented above, we can conclude that even within a single matrix (i.e., honey), the impact of temperature on polyphenols varies depending on the origin and type of honey.

## 7. Challenges in the Analysis of the Extractant

To develop an understanding of the impact of temperature on the extraction of polyphenols, the extractants must be critically analysed. The total phenolic content (TPC) and antioxidant activity (AOA) are the most commonly used measures. The TPC is usually calculated using the Folin–Ciocâlteu assay or by HPLC analysis. The AOA can be calculated using a variety of assays: ABTS (2,2-azinobis (3-ethyl-benzothiazoline-6-sulfonic acid)), DPPH (2,2-diphenyl-1-picrylhydrazyl), FRAP (ferric-reducing antioxidant power or ORAC (oxygen radical absorption capacity) assays [86]. Most studies have performed more than one antioxidant assay to analyse the extractant.

### 7.1. Overestimation of Total Phenolic Content by Folin–Ciocâlteu Assay

We need to develop a further understanding of how the TPC, as measured by the Folin–Ciocâlteu method, corresponds to the TPC measured by HPLC. It is difficult to measure TPC using HPLC, as a standard of each phenolic acid is needed to identify the peaks and quantify the area and, as a result, TPC calculated by HPLC is often much lower than that measured by the Folin–Ciocâlteu method [22,73].

As the Folin–Ciocâlteu method depends on the reducing power of phenolic hydroxyl groups to estimate the TPC, it accounts for all the phenols and their degraded products. This lack of specificity often results in an overestimation of TPC.

According to Vergara-Salinas et al. [47] the TPC calculated by Folin–Ciocâlteu increased with increasing temperature while the measurements of individual polyphenols and total polyphenols as calculated by HPLC suggested the opposite. The various polyphenols tested were hydroxyphenyl propanoic acids (HPPA), hydroxycinnamic acids, flavones, and flavanols/flavanones. Except for HPPA, the yield was highest at 100 °C and lowest at 200 °C for all compounds. HPPA and TPC calculated by the Folin–Ciocâlteu assay showed increased yield with increasing temperature. The total phenols calculated by the sum of areas of peaks in the HPLC chromatogram is lowest at 200 °C and highest at 100 °C, in contrast to the TPC measured by Folin–Ciocâlteu assay. At temperatures above 160 °C, water is able to solubilise even lignin and hemicellulose [47]. With the increase in solubility and hydrolytic reactions at high temperatures, these compounds may break down into phenolic acids, and the breakdown of lignocellulose may release not just phenolic materials but also reducing agents and sugars [54]. These additional compounds could also be detected by the Folin–Ciocâlteu assay causing further errors in the estimation of TPCs.

Mišan et al. [73] performed the quantification of phenolics (TPC) by HPLC and compared the result to the TPC values obtained by Folin–Ciocâlteu assay. The study included parsley, buckthorn, mint, caraway and birch extracts and found no significant correlation between the results of HPLC and the Folin–Ciocâlteu method. However, the difference between the results was not very high for parsley, mint and buckthorn. TPC in birch and parsley was overestimated by the Folin–Ciocâlteu method, while it underestimated caraway. This suggests the effectiveness of measuring TPC using the Folin assay varies depending on the source material.

Mandic et al. [87] found that the TPC and TFC calculated by the Folin–Ciocâlteu and HPLC methods is highly correlated (r = 0.90). However, the Folin–Ciocâlteu method resulted on average in a TPC 1.5–2.5 times higher than the HPLC results. Guendez et al. [88] found that TPC (HPLC) correlates to AOA with $r^2 = 0.628$ and TPC (Folin–Ciocâlteu) correlates to AOA with $r^2 = 0.649$. Once again, the TPC (HPLC) values are on average 2.9 times lower than TPC (Folin–Ciocâlteu) values, suggesting that other compounds present in the extracts that are accounted for in the TPC have little influence on the overall AOA.

### 7.2. Lack of Consensus in the TPC Correlation to AOA

While most of the literature did find a strong correlation between TPC and AOA [19, 20,28,57,87–89], this is not always the case. There are quite a few publications that did not show such a correlation [43,46–48,90–92].

Budrat and Shotipruk [48] extracted polyphenols using PLE (130–200 °C) and methanol/water extraction from bitter melon (65 °C). TPC increased with increasing temperature as did the specific polyphenols (catechin, gallic acid, gentisic acid and chlorogenic acid) that were studied, although gallic acid showed a slight decrease at 200 °C. As the AOA does not relate to TPC or the individual phenols here, the AOA must be exhibited by either phenolic compounds not tested for here or non-phenolic compounds such as vitamins and sugars that were decomposed at higher temperatures.

The highest AOA was observed at 150 °C and the lowest was at 200 °C, showing that higher extraction temperatures result in lower AOA. While the highest TPC was observed at 200 °C, it corresponds to the lowest AOA. However, even the lowest AOA observed at 200 °C in the PLE extraction is seven times higher than any of the traditional extractions performed (Soxhlet, methanol (at 65 °C) and water extraction (at 100 °C)).

Rodríguez-Meizoso et al. [46] performed a PLE extraction of polyphenols from dried oregano leaves at 25, 50, 100, 150 and 200 °C. The study showed that the temperatures did not significantly affect the TPC, although a drop was observed at 200 °C. The authors concluded that PLE does not, therefore, lead to the degradation or oxidation of phenolic compounds until 200 °C. However, the AOA increased at higher temperatures, suggesting that TPC is not correlated to AOA. The authors suggested that although the number of phenolic compounds is relatively constant, the variety and structure of the compounds may be changing in a way that increases the AOA.

The relationship between the structure of a polyphenol and its antioxidant capacity is not well understood. However, some studies have tried to establish a link between them. Benavente-García et al. [93] studied flavonoids in the citrus peels and established that 'the antioxidant capacity of any flavonoid will be determined by a combination of the O-dihydroxy structure in the B-ring, the 2,3-double bond in conjugation with a 4-oxo function and the presence of both hydroxyl groups in positions 3 and 5′.

## 8. Conclusions

To understand the behaviour of polyphenols during extraction at high temperatures, one needs to understand all the factors that affect this parameter. Thermal degradation of different types of polyphenols occurs at different temperatures, but it depends on the pre-treatment, solvent type, pH, treatment time, extraction environment and source of the material. The extraction temperature has a significant effect on the types of polyphenols being extracted. Further studies are needed to understand the role of specific phenolic acids and their antioxidant activity. The formation of new compounds (MRPs) at the high temperatures under which PLE is performed should also be investigated.

The lack of specificity of the Folin–Ciocâlteu assay for calculating TPC and the lack of understanding on how TPC relates to AOA makes it very difficult to establish a clear understanding of the reported conflicting effect of temperature on polyphenols. It is recommended that HPLC analysis of various phenolic acids should be performed along with TPC by Folin–Ciocâlteu assay to develop a better understanding of the extracts phenolic profile. The review concludes that thermal degradation alone does not explain the decrease in phenolic yield at temperatures above 90 °C, and all the factors discussed in the paper should be taken into account to understand the effect of temperature on polyphenols.

**Author Contributions:** Conceptualization, A.A. and M.F.; methodology, A.A. and M.F.; investigation, A.A.; writing—original draft preparation, A.A.; writing—review and editing, M.F.; supervision, M.F. All authors have read and agreed to the published version of the manuscript.

**Funding:** This research received no external funding.

**Informed Consent Statement:** Not applicable.

**Conflicts of Interest:** The authors declare no conflict of interest.

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
