# Peer review of "Effect of Temperatures on Polyphenols during Extraction"

_applsci, doi:10.3390/app12042107_

Round 1

Reviewer 1 Report

The manuscript submitted by Antony & Farid addresses an exciting topic of interest to anyone who studies polyphenols in food. The manuscript is well structured and presents relevant information. I suggest reviewing the following manuscripts that address the effect of heating on the content of polyphenols in a well-known matrix such as honey, and the other in a group of polyphenes of great importance such as anthocyanins (10.1016/j.foodchem.2021.130519; 10.1016/j.tifs.2021.01.049; 10.1016/j.foodchem.2018.07.012; 10.1016/j.foodchem.2017.08.054)

Author Response

Thank you for taking the time to review our manuscript. We appreciate your kind words and feedback. We have revised the manuscript and included various studies that specifically studied honey in “6. Variance of Source” section, lines 320-345. We have also expanded on the impact of heat on anthocyanins in the “2. Thermal degradation” section, lines 135-152. We have also reviewed the papers you suggested:

10.1016/j.foodchem.2021.130519: Now included in the manuscript, line 327-328

10.1016/j.tifs.2021.01.049: Now included in the manuscript, line 135-136 and line 207-210

10.1016/j.foodchem.2018.07.012: This study did not report the total phenolic content for each temperature tested, so I wasn’t able to isolate the thermal effect on the polyphenols from this study. Thus, it wasn’t added to the manuscript.

10.1016/j.foodchem.2017.08.054: This study did not test for total phenolic content or antioxidant activity, so it wasn’t added to the manuscript.

Once again, thank you very much for your valuable feedback, I hope the changes we have made have improved the manuscript.

Reviewer 2 Report

Authors have reviewed an interesting topic "Effect of temperatures on polyphenols during extraction". However there are a number of past studies which have been reviewed. The present study lacks the method evaluation and gap analysis. Further, there was also missing the productive direction or author suggestion in each methods of extraction where temperature affecting the polyphenols yield.  

Author Response

Thank you for taking the time to review our manuscript. We appreciate your feedback. As there are currently no review papers published that analyse the impact of temperature on the extraction of polyphenols, we hoped this manuscript would fill that research gap. We have added our suggestions to the “Conclusions” section in the revised manuscript to give productive direction as to what studies are needed in the field to better understand the thermal effects on polyphenols.

Once again, thank you very much for your valuable feedback, I hope the changes I have made have improved the manuscript.

Reviewer 3 Report

The manuscript is well-written and may be accepted after minor revisions. Here are revisions suggested 1.Effect of Enzymes (3.2) should be expanded. 2.Some references are not per MDPI format. They may be corrected. 3. The first reference may be replaced by recent reference Zia-Ul-Haq, M. Historical and introductory aspects of carotenoids. In Carotenoids: Structure and Function in the Human Body; Zia-Ul-Haq, M., Dewanjee, S., Riaz, M., Eds.; Springer: Cham, Switzerland, 2021; pp. 1–42.

Author Response

Thank you for taking the time to review our manuscript. We appreciate your kind words and feedback. Here are the modifications made to the manuscript as per your suggestions:

  1. Effect of Enzymes (3.2) should be expanded
    • The enzymes section (3.2) has been rewritten and expanded (lines 207-224)
  2. Some references are not per MDPI format. They may be corrected.
    • Every reference has been double-checked and edited to fit the MDPI format
  3. The first reference may be replaced by a recent reference
    • The suggested reference has been added to the manuscript (citation 2, line 37)

Once again, thank you very much for your valuable feedback, we hope the changes we have made have improved the manuscript.